



# A continuous 6000 year age depth relationship for the remainder of the Weißseespitze summit glacier based on $^{39}$Ar and $^{14}$C dating

David Wachs[1,2], Azzurra Spagnesi[3,6], Pascal Bohleber[4,5], Andrea Fischer[6], Martin Stocker-Waldhuber[6], Alexander Junkermann[2], Carl Kindermann[1], Linus Langenbacher[1], Niclas Mandaric[2], Joshua Marks[1,7], Florian Meienburg[1,2], Theo Jenk[8], Markus K. Oberthaler[2], Werner Aeschbach[1,9]

[1]Institute of Environmental Physics, Heidelberg University, Heidelberg, Germany
[2]Kirchhoff-Institute for Physics, Heidelberg University, Heidelberg, Germany
[3]Department of Environmental Sciences, Informatics and Statistics, Ca' Foscari University of Venice, Venice, Italy
[4]Alfred Wegener Institute Helmholtz Centre for Polar and Marine Research, Bremerhaven, Germany
[5]Goethe University Frankfurt am Main, Frankfurt am Main, Germany
[6]Institute for Interdisciplinary Mountain Research of the Austrian Academy of Sciences, Innsbruck, Austria
[7]Helmholtz Centre for Environmental Research - UFZ, Magdeburg, Germany
[8]Paul Scherrer Institute, Villigen, Switzerland
[9]Heidelberg Center for the Environment, Heidelberg University, Heidelberg, Germany

*Correspondence to*: David Wachs (David.wachs@iup.uni-heidelberg.de)

**Abstract.**

Associated with ongoing global warming, prolonged periods of negative mass balance affect even Alpine glaciers in high summit regions, which are also prime candidates for paleoclimate-related ice core studies. This greatly complicates the already challenging task of establishing an age-depth relationship where now both, the age at depth and at the surface is an unknown. Radiometric ice dating methods are an important key to tackle this challenge. This study presents a comprehensive age-depth profile of the summit glacier of Weißseespitze (WSS, 3500 m a.s.l.) in the Austrian Alps, utilizing a novel combination of radiometric dating methods – $^{39}$Ar and $^{14}$C. Ice cores from drilling campaigns conducted in 2019, 2023, and 2024 were analyzed to overcome challenges posed by extensive ice loss and surface melting that limit traditional dating techniques. All $^{39}$Ar samples were measured using atom trap trace analysis (ATTA). Surface mass balance (SMB) data since 2019 were used to align core depths across years, and all samples were referenced to height above bedrock to standardize comparisons.

Age modeling using least squares fitting and Monte Carlo sampling was performed for three glaciological models: Nye, Raymond, and a two-parameter (2p) model to test their applicability. The 2p model provided the best fit ($\chi^2_{red}$ = 0.4), closely matching the data and providing a continuous age-depth scale. The model yielded a mean accumulation rate of 0.53 m w.e. a$^{-1}$ (1σ range: 0.39–0.63 m w.e. a$^{-1}$) and a thinning parameter p = 0.92 (1σ: 0.81–0.97), the former agreeing with current accumulation estimates.

The results show that the surface ice dates back approximately 400 a, emphasizing the extent of recent ice loss. Apart from this, the continuous age-depth relation shows no sign of prolonged periods of mass loss at WSS within the 6000 a glaciation history prior to today.



This work underscores the utility of [39]Ar dating in alpine glaciology, enabling precise reconstruction of age-depth relationships even under advanced glacial retreat and enhancing our understanding of Holocene climate history in the Eastern Alps.

## 1 Introduction

Glaciers constitute one of the most valuable climate archives on Earth. They do not only record climate variability in their size and location, under specific conditions they can also archive atmospheric signals in their ice matrix and the enclosed bubbles. While polar ice sheets provide long records on global climate signals (e.g. EPICA community members, 2004), non-polar glaciers record more regional signals. However, as they are located in closer proximity to human settlements, they are also able to record the anthropogenic impact on a regional scale. High elevation glaciers can hereby provide continuous signals

covering large parts of the Holocene (Bohleber, 2019) . However, with the current mass loss of Alpine glacier ice an almost complete disappearance of glaciers from most parts of the Alps is expected within this century (Hartl et al., 2024). Therefore, there is particular urgency to retrieve the ice records before they are irrevocably gone.

A crucial step toward a meaningful interpretation of any signal from an ice core is the construction of a precise and robust chronology. For Alpine glaciers the common approach via layer counting often fails because of complex wind-dominated

snow deposition and rapid layer thinning with depth. Furthermore, the prolonged negative surface mass balance associated with ongoing warming adds a new level of complexity: the melting of the surface leading to unknown surface ages. Under these circumstances, radiometric dating methods are the only way to determine a chronology for an ice core (e.g. Festi et al., 2021). With the current persistent mass loss, surface ages can be in the range of several centuries. In this age range on the one hand traditionally applied tracers like $^3$H and $^{210}$Pb fail due to their short half-lives and on the other hand $^{14}$C dating is hampered

due to ambiguities in the calibration curve and the low carbon content of ice (Ramsey, 1995; Reimer et al., 2013). To cover the intermediate age range that is present in the upper layers of most Alpine ice caps, the radiometric dating tracer $^{39}$Ar is the optimal candidate. With a half-life of $268 \pm 3$ (Golovko, 2023) it spans an age range of roughly 50-1000 years. After a first application to ice samples by Feng et al. (2019) and to an ice core by Ritterbusch et al. (2022) it is now available as a routine measurement for glacier ice dating. A recent example for the value of $^{39}$Ar in glacier dating in combination with micro-

radiocarbon ages has been provided by Legrand et al. (2025).

In this work, the $^{39}$Ar dating tracer is applied to the upper part of the summit glacier of the Weißseespitze (WSS, 3500 m a.s.l.), Austria. The glacier has a dome shaped geometry, a rare feature for Alpine glaciers, which is favorable for the preservation of old ice. In combination with the $^{14}$C data presented in Bohleber et al. (2020), the complete remaining ice profile is now covered by radiometrically measured ages. By applying different models to the age-depth profile, a suitable model is determined (2p

model, Bolzan, 1985; Thompson et al., 1990) and a continuous time scale is established despite 80 % of the original glacier already having disappeared.



The time scale reaches from a 2019 surface age of almost 400 a back to about 6000 a at the base of the glacier. It is an example of the relevance of a combination of $^{39}$Ar and $^{14}$C to cover the age profile of a glacier in times where surface ages have become unknown due to mass loss, calling for radiometric dating methods to fill the gap.

**2 Glaciological setting of the WSS summit glacier**

The WSS summit glacier (3500 m a.s.l.) marks the highest point of the Gepatschferner glacier. Its dome-shaped geometry makes it particularly interesting as it offers conditions of very slow ice flow and therefore good preservation conditions for old ice. Figure 1 shows an overview of the study site as well as photos depicting the summit glacier (also see Figure S1 in the SI for further aerial photos of the summit region). Measurements of borehole temperatures indicate that the ice is still frozen to

the bedrock (Bohleber et al., 2020; Fischer et al., 2022; Stocker-Waldhuber et al., 2022b). The ice thickness of the summit glacier was measured to be 10 m at the ice divide by Bohleber et al. (2020) and is rapidly declining (Fischer et al., 2022; Hartl et al., 2025; Stocker-Waldhuber et al., 2022a). Fischer et al. (2022) provides a detailed discussion of historical sources and states a maximum elevation of the ice cap at the end of the 19th century at about 3534 m a.s.l., translating to an ice cap thickness of about 47 m. Current average surface mass loss rates are projected to result in a complete loss of the summit glacier within

the next 10-20 a. Recent ice thickness measurements from 2025 revealed a remaining ice cap thickness of less than 6 m close to the drilling site.

Furthermore, also the basal layers will soon warm above 0°C and cause the loss of the oldest ice remains frozen to the bedrock, which constitute an alternative climate signal for periods of low ice cover in the Alps (Bohleber et al., 2020).





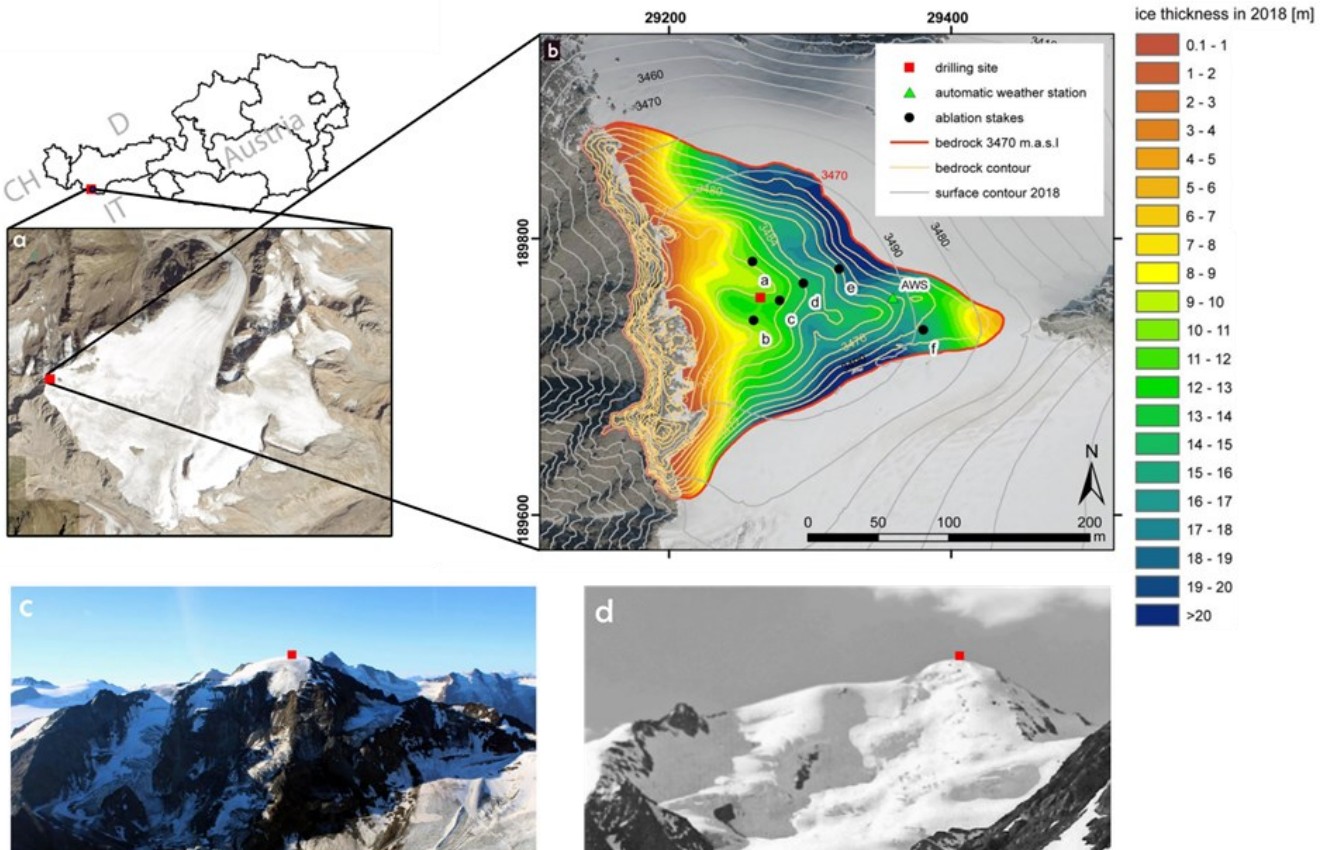

**Figure 1: Drilling site at the summit glacier of Weißseespitze (red square), source: maps.tirol.gv.at. (a). Bedrock topography from the GPR measurements and calculated ice thickness in 2018 (from Fischer et al., 2022) (b). View of the Weißseespitze ice cap in 2019 (c) and 1930 (d). Adapted from Fischer et al. (2022) and Spagnesi et al. (2023).**

## 2.1 Surface Mass Balance and depth-scale alignment

Combining the data sets from ice samples retrieved in different years allows for a complete coverage of the profile of the ice sheet on WSS by measured ages. However, as the cores taken in 2023 and 2024 were not drilled to bedrock and no common horizon was visible in the ice, the depth scale has to be established from the annual surface offsets. For this, the annual SMB, listed in Table 1, is considered. As indicated by Fischer et al. (2022) and Hartl et al. (2025), the ablation can vary strongly across the summit region where differences between ablation stakes of up to 75 cm were seen. Furthermore, despite using a precise GPS to find the drilling location from former years, spatial deviations of up to 10 m are possible. Figure 1 (b) shows that this can lead to depth topography changes of up to about 2 m. However, the effect is probably significantly smoothed out by the snow and ice build-up and effects like wind erosion. A well-visible dust layer that melted out quite homogeneously




across the summit region in 2022 (see Fig. S2 in SI), rather points towards spatial layer height variations around 10-20 cm. In consequence, depth values of years other than 2019 are shifted according to the values in Table 1 and a conservative error of

1 m is estimated.

**Table 1: Annual SMB taken from Stocker-Waldhuber et al. (2025) and Hartl et al. (2025). The values indicate the mass loss during the summer of the respective year. The sampling at WSS always took place in the spring before the mass loss of the respective year.**

| Year [CE] | 2019 | 2020 | 2021 | 2022 | 2023 |
|---|---|---|---|---|---|
| Surface Mass Balance [m i.e.a$^{-1}$] | -0.22 | -0.00 | -0.16 | -1.29 | -0.16 |

Furthermore, as depth always refers to a reference surface, which in this setting is changing over time, everything is discussed

in terms of height over bedrock in the following, instead of depth below some annual surface. The conversion was done with the 2019 thickness of the ice sheet of 10.2 m (from the logging data of the 2019 sampling campaign, length of core 1) and the surface mass balance listed in Table 1. With this measure, a more general metric is established, which can more easily be applied to potential future ice cores that are drilled to bedrock. Nonetheless, because the wording is otherwise uncommon, the profile of ages throughout the ice column is still referred to as *age-depth profile*.

**2.2 Last steady state**

All the models which are used in the following to fit the data describe glaciers in a steady state. This means that the conditions at the *last steady state* (LSS) at WSS have to be estimated and the models have to be fitted accordingly. The basic Sorge's law of glaciology states that the age-depth profile of an ice sheet remains constant under steady-state conditions. From Fischer et al. (2022), WSS is estimated to have experienced its LSS around the year $1914 \pm 50$ at an ice thickness of $47 \pm 10$ m. The

timing of the LSS was chosen as the last time interval which showed no significant ice loss. However, accounting for the low accuracy of the early data stemming mostly from old maps and aerial photos, a conservative error of 50 a is chosen. This way, the LSS is estimated to lie between the Little Ice Age (LIA) maximum around 1864 and the beginning of in-situ SMB measurements in 1969. With these estimates, the measured ages can be shifted to the time of the last steady state to fit the models before translating the age profiles back to 2019.

**3 Methods**

**3.1 Sample retrieval and processing**

In March 2019, two parallel ice cores were drilled at the ice divide with nearly flat bed conditions. Analysis for micro-radiocarbon ice dating was performed on discrete samples from one of the cores at the Paul Scherrer Institute (PSI), and $^3$H was measured on three additional shallow cores from the same location at Seibersdorf Laboratories (Bohleber et al., 2020).

The $^3$H analysis showed no sign of the elevated levels from after the 1960s, indicating surface ages beyond 60 a. The




radiocarbon analysis, conducted for samples from the lowest six meters above bedrock, revealed an age profile that increases with depth reaching a maximum calibrated ¹⁴C age of approximately 6000 a at the bottom and decreasing to below 1000 a at approximately 5 m above bedrock. A comprehensive account of the data set and the sample processing can be found in Bohleber et al. (2020).

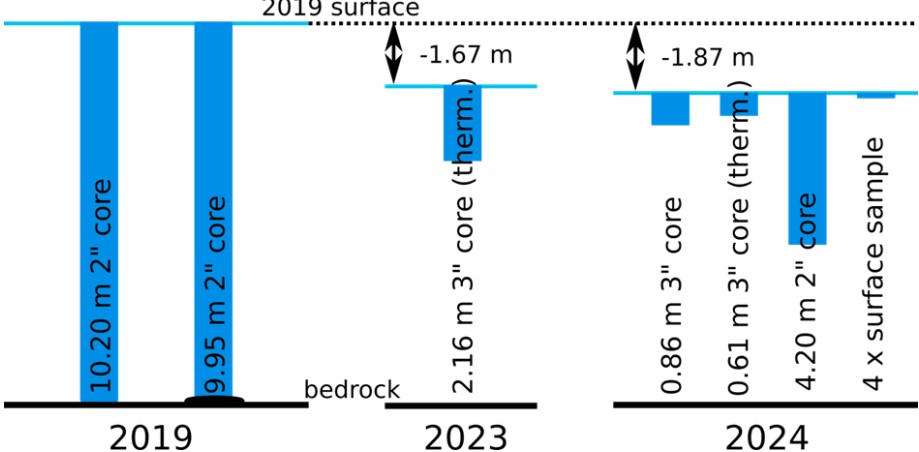


**Figure 2: Overview of the ice core drilling of different years. Indicated are the length of the samples, the type of sample (retrieval method) as well as the changes in surface height above bedrock between the sampling years.**

To better constrain the ages at the top of the ice column, several sampling campaigns have been conducted in recent years to obtain ice for ³⁹Ar dating. In June 2023, a first attempt to retrieve a new core at the summit of WSS was conducted, however

retrieving only 2.16 m with a thermal 3" drill due to equipment failure.

A second attempt was conducted in May 2024, which again experienced a failure of the drilling equipment. Still, four surface samples and several cores (both, thermally and mechanically drilled) were taken. An overview of the sampling of the different years is given in Figure 2.

For the 2023 and 2024 campaigns, the ice was packed into insulation boxes on site and transported to the cold-room facilities

(−20°C) at the Institute of Environmental Physics, Heidelberg. Here, the ice was decontaminated by cutting off the outer 2 mm and the gas in the air bubbles extracted and purified following the previously detailed procedure by Feng et al. (2019) and Arck et al. (2025). The combined ice pieces were then transferred into a 12.6 l stainless steel container. After evacuating the container, the ice was melted and the released gas captured on liquid nitrogen cooled activated charcoal traps. To purify the contained argon, all reactive gases were removed using two getters at 900°C and room temperature in series (Arck et al., 2025).

The remaining >98% pure argon gas was collected on an activated charcoal trap. Using this method, between 0.4 to 1 ml of pure argon was extracted from 2.5 to 4.6 kg of sample ice.



### 3.2 $^{39}$Ar measurement and analysis

$^{39}$Ar measurements were conducted in the ATTA laboratory at the Kirchhoff-Institute for Physics, Heidelberg. The atom-optical ATTA method enables the cooling and trapping of individual atoms in order to count them (for details see Ebser et al., 2018 and Lu et al., 2014) . $^{39}$Ar is measured as an isotopic abundance ($^{39}$Ar/Ar) and reported in percent modern argon (pmAr), which is the sample abundance normalized to the modern atmospheric abundance ($R_A$). By comparing the obtained count rate to reference measurements with known isotopic abundance, the $^{39}$Ar abundance in the sample can be inferred. With usual count rates of around 6 $^{39}$Ar atoms h$^{-1}$, sample measurements were conducted in a gas recycling mode over 20 h to ensure high enough statistics (Ebser et al., 2018 and Feng et al., 2019). Reference measurements were done with an enriched reference gas (10 $R_A$), which was measured for 2 h to obtain similar statistics to the samples. Furthermore, to account for background counts in the apparatus, blank measurements are conducted on a weekly basis and incorporated into the data analysis.

The newly developed data analysis applied in this work incorporates all reference and blank measurements over a measurement period with an unchanged, stably operating apparatus, typically spanning one to several weeks. The data analysis employs the underlying Poissonian statistics of atom counting to fit the time between atom detection events with the exponential distribution

$$p(\Delta t; r) = e^{-r\Delta t}, \tag{1}$$

where $r \left[\frac{1}{s}\right]$ denotes a rate parameter which is a function of the abundance in the sample, the performance of the apparatus (determined by the reference measurements), and of $\Delta t$ itself due to contamination build-up in the system. The rate of the latter is determined by blank measurements. The contamination build-up originates from the recirculation of the sample gas in the apparatus allowing for an accumulation of degassing $^{39}$Ar from the vacuum parts. Furthermore, the data evaluation routine performs a quality check of the measurement by testing for statistical outliers in the atom detection frequency, rejecting irregular measurements due to potential instabilities in the measurement.

The resulting $^{39}$Ar abundance is then translated to an age using the $^{39}$Ar half-life of $268 \pm 3$ a and a reconstructed atmospheric input curve similar to Gu et al. (2021).

## 4 Results

### 4.1 Radiometric age dating

Table 2 gives an overview of all measured samples. The $^{39}$Ar samples are from the 2023 and 2024 sampling campaigns. All ages are already corrected for the varying atmospheric input function. Two samples, indicated with an asterisk, did not provide enough gas for a measurement and had to be diluted with $^{39}$Ar-free Ar. The presented abundances and ages are already corrected for this mixing.

The $^{14}$C samples were measured on an ice core from 2019 and all except one are presented in the publication by Bohleber et al. (2020). The additional data point of the same core, WSS_Run15, was also measured at PSI in 2024 and added to the dataset.



Figure 3 shows a depth profile of all $^{39}$Ar and $^{14}$C ages. The colors indicate the different years of sampling. Furthermore, data points not considered in the following are indicated in red. This selection was due to several quality checks. The samples WSS24_SurfaceA and WSS24_SurfaceB were discarded due to atmospheric contamination originating from incomplete cleaning of the ice prior to the gas extraction. Sample WSS24_Core3_Sample3 consisted of very brittle ice and was therefore

185    likely to incorporate modern atmospheric air. Finally, sample WSS24_SurfaceC did not pass the final data analysis due to unstable measurement conditions.

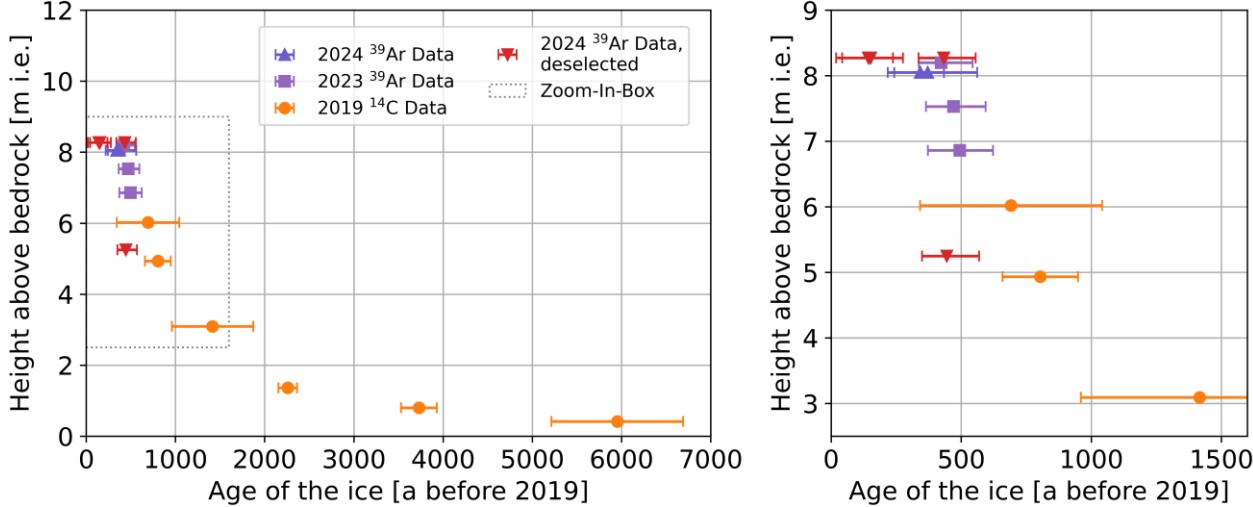

**Figure 3: All $^{39}$Ar and $^{14}$C data points from WSS plotted as age vs. height above bedrock (in meters ice equivalent (m i.e.)), depth intervals of the data points are not included for better visibility. On the left, the full profile can be seen, on the right a close-up view**

190    **of the upper (younger) part is displayed. Data from different years are indicated in different colors. The red data points did not pass the quality checks as discussed in the main text and will be omitted from further analysis.**

**Table 2: Overview of all samples from the 2019, 2023 and 2024 ice cores which were analyzed for $^{39}$Ar and $^{14}$C. The $^{39}$Ar measurements were conducted in Heidelberg, $^{14}$C measurements originate from Bohleber et al. (2020), except for sample 15, which was analyzed in 2024. The height above bedrock values are calculated as indicated in Sect. 2.1 Surface Mass Balance and depth-**

195    **scale alignment. Ages are given as input curve corrected values before 2019. Argon values with an \* indicate samples which were diluted with $^{39}$Ar dead argon prior to the measurement to provide enough gas. The shown pmAr and age values are already corrected for this dilution. Italicized samples were not considered in the final analysis due to reasons discussed in Sect. 4.1.**

| Sample | Type | Height above bedrock [m] | Length [m] | Weight [kg] | Argon [ml] | $^{39}$Ar/$^{14}$C [pmAr/ pmC] | Corrected age [a b. 2019] |
|---|---|---|---|---|---|---|---|
| WSS23_Run12upper3 | $^{39}$Ar | 8.20 | 0.66 | 2.85 | 0.84 | $38^{+11}_{-10}$ | $393^{+121}_{-85}$ |
| WSS23_Runlower34 | $^{39}$Ar | 7.53 | 0.66 | 3.09 | 0.65 | $34^{+11}_{-10}$ | $440^{+124}_{-106}$ |
| WSS23_Run5 | $^{39}$Ar | 6.86 | 0.66 | 2.63 | 0.53 | $32^{+12}_{-10}$ | $465^{+128}_{-122}$ |
| *WSS24_SurfaceA* | $^{39}$Ar | 8.27 | 0.10 | 2.50 | 0.39* | $78^{+26}_{-22}$ | $113^{+133}_{-124}$ |
| *WSS24_SurfaceB* | $^{39}$Ar | 8.27 | 0.15 | 4.56 | 0.99 | $76^{+20}_{-15}$ | $122^{+86}_{-109}$ |
| *WSS24_SurfaceC* | $^{39}$Ar | 8.27 | 0.13 | 3.76 | 0.77 | $36^{+12}_{-10}$ | $403^{+123}_{-97}$ |




| | | | | | | | |
|---|---|---|---|---|---|---|---|
| WSS24_Core1_Sample1 (mechanical) | $^{39}$Ar | 8.05 | 0.61 | 2.55 | 0.64 | $48^{+13}_{-11}$ | $313^{+90}_{-100}$ |
| WSS24_Core2_Sample1 (thermal) | $^{39}$Ar | 8.05 | 0.61 | 2.47 | 0.52* | $43^{+22}_{-17}$ | $342^{+190}_{-153}$ |
| *WSS24_Core3_Sample3 (handheld mechanical)* | $^{39}$Ar | 5.25 | 2.11 | 4.40 | 0.72 | $36^{+11}_{-10}$ | $415^{+124}_{-95}$ |
| WSS19_Run7 | $^{14}$C | 6.02 | 0.70 | 0.40 | - | $93.0 \pm 4.6$ | $692 \pm 350$ |
| WSS19_Run9 | $^{14}$C | 4.94 | 0.55 | 0.25 | - | $91.0 \pm 1.9$ | $804 \pm 145$ |
| WSS19_Run12 | $^{14}$C | 3.10 | 0.59 | 0.44 | - | $84.7 \pm 4.6$ | $1416 \pm 456$ |
| WSS19_Run15 | $^{14}$C | 1.36 | 0.65 | 0.83 | - | $76.1 \pm 0.8$ | $2259 \pm 104$ |
| WSS19_Run16 | $^{14}$C | 0.80 | 0.48 | 0.30 | - | $65.7 \pm 1.3$ | $3730 \pm 200$ |
| WSS19_Run17-1 | $^{14}$C | 0.42 | 0.56 | 0.17 | - | $52.9 \pm 4.1$ | $5953 \pm 739$ |

## 5 Discussion

### 5.1 Age models

Three models were fitted by least squares (Python Scipy curve_fit, Virtanen et al., 2020) to the combined $^{39}$Ar and $^{14}$C data to test their applicability to the setting. In the following, all models are stated in terms of the age $t$ in dependance of the height above bedrock $h$. The relation of $h$ to the otherwise employed depth coordinate $z$ is given by $h = H - z$, with $H$ being the ice thickness in the steady state.

The first applied model is given by the Nye timescale (Nye, 1957, 1963), described by

$$t(h) = \frac{H}{b} \ln\left(\frac{h}{H}\right),$$ (2)

where the model parameter $b$ denotes the accumulation rate.

The second applied model was proposed by Raymond (1983) and is conceptualized for an ice-divide type setting:

$$t(h) = -\frac{H}{b}\left(1 - \frac{H}{h}\right).$$ (3)

The third model fitted to the data is the 2-parameter (2p) model by Bolzan (1985) and Thompson et al. (1990), which was successfully applied to an Alpine ice core from Colle Gnifetti by Jenk et al. (2009) and is given by:

$$t(h) = -\frac{H}{bp}\left(1 - \left(\frac{H}{h}\right)^p\right).$$ (4)

Effectively, the 2-p model adds a thinning parameter $p$ to the Raymond model that was originally added to account for a non-isothermal polar ice sheet (Bolzan, 1985). For $p = 1$, the 2-p model equals the Raymond model.





## 5.2 Monte-Carlo sampling and fitting

To obtain sensible age error estimates for the models, a Monte-Carlo (MC) sampling based approach was developed. Both the $^{39}$Ar as well as the $^{14}$C age data are given by non-symmetric probability distributions. Their non-Gaussian shape stems from the non-linear relationship between isotope abundance and age due to the exponential radioactive decay law as well as the varying atmospheric input functions of both tracers, and, in the case of $^{39}$Ar, the underlying counting statistics of the $^{39}$Ar measurement. To consider the full information of the distributions, an approach via MC sampling from the age distributions was taken. For each realization the distribution of each sample age was sampled and the obtained ensemble fitted with the model. The fitting of each realization was done by least squares fitting. While the best fit was obtained from fitting the measured values, the age uncertainties of the models at each depth were taken as the respective quantile of the MC fit realizations at each depth. Thus, the full complexity of the distributions for each data point can be accounted for in the fitting procedure. This was done for the data at hand with a sampling number of $n_{MC} = 10^5$.

Additional to the sampling from the age distributions of each measurement, the depth uncertainty of the $^{39}$Ar measurements derived from the depth scale alignment (see Sect. 2.1 Surface Mass Balance and depth-scale alignment) with an error of 1 m as well as the timing and thickness uncertainties for the LSS ($1914 \pm 50$, $47 \pm 10$ m, see Sect. 2.2 Last steady state) of the glacier were MC sampled as Gaussian distributed values with the indicated standard deviations. The following analysis is conducted for height in meters ice equivalent (m i.e.), however no assumptions have been made regarding the firn thickness at the LSS, which would introduce a small thickness reduction in units of m i.e due to the low firn density. Exemplary model fits showed no mentionable difference (few permille) in the age-depth relation for firn layers around 20 m.

With this approach, the non-linear interplay of the different parameters and their uncertainties are considered in the final age uncertainties by fitting statistically distributed data ensembles which contain all uncertainties in their distributions.

There is a distinct difference between the two employed tracers. While $^{14}$C dates the microscopic carbon deposits in the ice and thus the actual ice matrix, $^{39}$Ar dates the gas trapped in the ice. The time interval between snowfall and the pore close-off further down in the firn column causes an age difference between dating tracers in the ice matrix and the gas, referred to as Δage. To correct for this difference, 30 a of Δage were added to the $^{39}$Ar ages in the fitting process, an estimate based on the findings at nearby glaciers by Fest et al. (2021) and Gabrielli et al. (2016).

## 5.3 Difference between models

Table 3 lists the fitting results for the three models. From its last column, displaying the reduced chi-squared ($\chi^2_{\text{red}}$) value, it is apparent that the Nye model is not able to capture the age profile of this ice cap. The model does not capture the thinning of the annual layers in the deeper parts as is also apparent from the visualization in Figure 4. The two other models, conceptualized for an ice divide setting, are better suited to describe the age-depth relationship at WSS because of the dome shaped summit glacier. Due to its flat bed and therefore low flow conditions, it resembles the setting of an ice divide. The $\chi^2_{\text{red}}$ for the



Raymond and 2p model are below 1 and thus demonstrate the good applicability of the models. As the 2p model shows the better fit quality it will be discussed in more detail in the following.

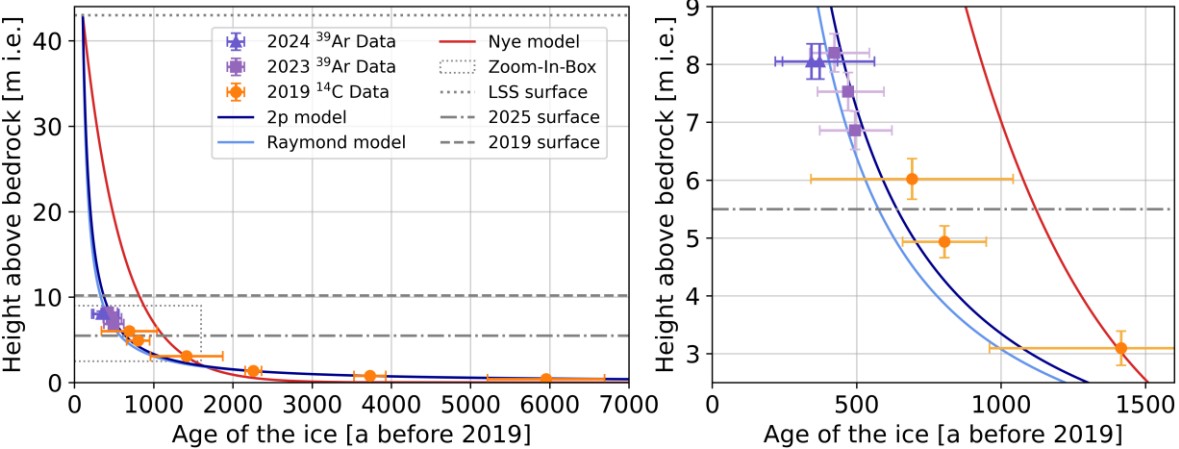

**Figure 4: Plot of the best fits of the different models (the respective parameters can be found in Table 3) and age data points vs. their height above bedrock (in meters ice equivalent). The maximum of the height axis is set to the last steady state (LSS) from 1914 (47 m ice thickness) and thus the full age-depth profile during the LSS is shown. The left side shows the full profile, the right side shows a close-up view of the younger section. The depth error bars indicate the length of the sample, not the depth uncertainty.**

**Table 3: Parameters for the best fit of the measured data and the 1σ boundaries from the MC fitting. Accumulation rate values are converted to meter water equivalent (m w.e.) by multiplying with an ice density of 900 kg m⁻³ (Festi et al., 2021).**

| Model | Accumulation rate, **best fit**, lower 1σ, upper 1σ bound [m w.e.a⁻¹] | Thinning parameter, **best fit**, lower 1σ, upper 1σ bound | $\chi^2_{red}$ |
|---|---|---|---|
| Nye model | **0.089**, 0.083, 0.095 | | 25.14 |
| Raymond model | **0.67**, 0.63, 0.72 | | 0.54 |
| 2p model | **0.53**, 0.39, 0.63 | **0.92**, 0.81, 0.97 | 0.4 |

**5.4 Surface ages**

In the case of a glacier which is losing significant amounts of ice on its surface every year, each surface age needs an indication of the year it refers to. For the years 2023 and 2024, the uppermost $^{39}$Ar measurements show ages of 300-400 a (compare Table 2). However, the samples consist of more than 60 cm long core pieces, so that a precise surface age cannot be given. Unfortunately, all three measurements from more shallow surface blocks had to be discarded (compare Sect. 4.1 Radiometric age dating





From the 2p model, the surface age for 2019, at 10.2 m above bedrock, is $371^{+79}_{-55}$ a, for 2023 $434^{+91}_{-64}$ a, and for 2024 $441^{+92}_{-65}$ a. The surface loss between 2019 and 2023 amounted to 1.67 m, while between 2023 and 2024 only 16 cm were lost.

These ages are clearly pre-industrial and show how far the prolonged mass loss is diminishing ice core records in Alpine regions and that [39]Ar was the only tracer to accurately constrain the surface age of this glacier. While tracers like [3]H or [210]Pb can determine young surface ages and thus complement [14]C ages from lower layers, this is often not possible anymore with the progressing melting of glaciers in the Alps. Without reliable surface ages, however, even if e.g. layer counting is possible, it can only provide a floating age-depth relationship with limited informative value.

The presented data set shows an example of the reconstruction of the age-depth relationship from a glacier of which only the lower ~20% are remaining, a setting which will become more frequent in the coming years. The combination of [39]Ar and [14]C data allows to fully cover of the important age ranges.

Also depicted in Figure 4 is the 2025 spring glacier surface after the 2024 summer melt, leaving the ice cap at a thickness of less than 6 m and emphasizing the rapid decline of the glacier.

## 5.5 Accumulation rate

One of the parameters fitted by the 2p model is the accumulation rate. However, the interpretation of the result is not trivial, as data from 6000 a is fitted to obtain this value. Due to the changing climatic history of the region and probably also changing weather patterns it cannot be expected that this parameter remained constant over time. Therefore, the result should be understood as an accumulation rate type parameter that gives a rough estimate of the average accumulation rate over the last six millennia. The 2p model results in a best fit value of b = 0.53 m w. e. a$^{-1}$, with 68.3% of the MC realizations resulting in

values between 0.39 and 0.63 m w. e. a$^{-1}$ (1σ interval). Within the 1σ interval this agrees with the value of 0.59 m w.e. which Fischer et al. (2022) gives for the total precipitation in months of snow accumulation for the years 1800 to 2014 (based on the HISTALP data from Auer et al., 2007).

The other fitted parameter, the thinning parameter p, results in a value of p = 0.92, with 68.3% of the MC realizations resulting in values between 0.81 and 0.97. This parameter was introduced by Bolzan (1985) to account for non-isothermal conditions in

the ice sheet, with p = 1 being the result for isothermal conditions (Raymond model). It is therefore not surprising to obtain a value close to 1 for a thin Alpine glacier, as was also found in the study of Jenk et al. (2009) for Colle Gnifetti (p = 0.867 ± 0.048).

## 5.6 Age-depth profile

With the [39]Ar data points completing the coverage of the whole ice column with measured ages and a suitable model to fit the

data, the age-depth profile of the WSS summit glacier is well constrained. In Figure 5 the resulting best fit of the 2p model and its uncertainties are displayed in an overview plot as well as a residual plot. The uncertainty ranges of the model (for $1\sigma$, $2\sigma$, and $3\sigma$ shaded in different shades of blue) are computed as the respective quantiles from the distribution of the MC model





fit realizations at each depth and therefore incorporate all measurement and LSS uncertainties. Figure S3 (in supplementary information) gives an overview of the distribution of the fit parameters for all fit realizations with indicated best fit parameters.

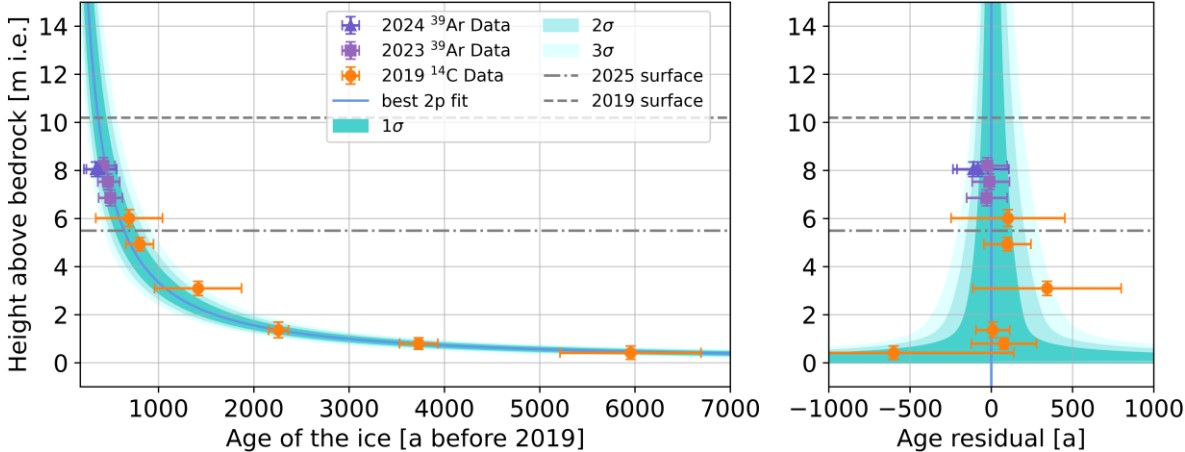


**Figure 5: Plot of the age-depth relationship from the 2p model on the left. On the right, a residual plot of the model and data points for better visualization. The shaded areas indicate the model age uncertainties obtained from the 1σ, 2σ, and 3σ values from the MC fitting. The depth error bars indicate the length of the sample, not the depth uncertainty.**

Considering the large amount of ice loss observed at WSS over the past decades, the question arises if the WSS summit has

experienced similar periods of strong mass loss during the ~6000 a recorded in the glacier ice. At least for the past 2000 a, a hiatus in the order of 400 annual layers seems to be unlikely given the good fit of the model to the data with errors of measurements and fit well below 400 a. For deeper layers, this is harder to constrain as the absolute errors of the [14]C ages increase. Overall, melt events erasing several hundred years cannot be excluded for times > 2000 a BP, however indications for such events are absent in the dataset.

As a further analysis of the applicability of the model, it was fitted separately to the upper six sample ages and the lower six sample ages. The results for the accumulation rate parameters do not deviate significantly from the original value: $b_{upper}$ = 0.71 (1σ interval: 0.46 to 0.91) [m w. e. a$^{-1}$] and $b_{lower}$ = 0.46 (1σ interval: 0.34 to 0.62) [m w. e. a$^{-1}$] (original value: 0.53 m w. e. a$^{-1}$). The approach of fitting the entire age profile with one of the applied age-depth models makes the simplified assumption of a constant accumulation rate over the entire time interval recorded in the ice column. Showing that the accumulation rate

derived by the fit parameter of the upper and the lower profile are consistent thus substantiates the validity of the approach.

In summary, this study shows the great value that [39]Ar dating has for alpine ice core dating. On the one hand, it provides surface ages where they become increasingly old thereby surpassing the range of traditional dating tracers. It can, on the other hand, also complete age-depth profiles where surface ages might still be acquired, but an obvious age data gap is present between the young age tracer results and the old ones as seen for example in the work of Jenk et al. (2009) and Gabrielli et al.

(2016). Legrand et al. (2025) for the first time demonstrated the smooth closure of such a gap with [39]Ar in a long ice core from



the high elevation summit glacier at Dôme du Goûter. For the study at hand, the $^{39}$Ar tracer was the only way to obtain precise age constraints for the upper half of the ice core and in combination with $^{14}$C ages results in a well-covered age-depth profile.

## 5.7 Sensitivity analysis with newly proposed $^{39}$Ar half-life

In a recent (not yet peer-reviewed) publication by the DEAP Collaboration (2025), the $^{39}$Ar half-life was redetermined by the

direct observation of the decay curve. The proposed value for the half-life is $302 \pm 8_{stat} \pm 6_{sys}$ a. This is a significant relative increase of 13% from the former value by Golovko (2023) and would affect all $^{39}$Ar ages. To estimate the effect of this change, the 2p model was fitted to the updated $^{39}$Ar ages together with the original $^{14}$C ages. The updated $^{39}$Ar ages were calculated also considering an atmospheric input curve corrected for the new half-life. The new ages are systematically older but lie within the uncertainties of the original age values stated above.

The resulting fit parameter values are: b = 0.48 (1σ interval: 0.35 to 0.57) [m w. e. a$^{-1}$] and p = 0.88 (1σ interval: 0.77 to 0.94) and thus do not change significantly with the new proposed half-life. The $\chi^2_{red}$ of the 2p model fit reduces to 0.26 due to the slight increase of all $^{39}$Ar age values (by 60-70 a for the ages around 400 a), which causes the $^{39}$Ar ages to agree better with the trend of the upper $^{14}$C ages (compare Figure 5).

## 5.8 Different sampling methods

During the 2023 and 2024 sampling campaigns, different sampling techniques were employed to retrieve the ice. Surface samples were taken by chain saw and ice cores were drilled mechanically and thermally (compare Table 2). Unfortunately, all surface samples had to be discarded from the data set due to different quality checks (see above). However, both mechanically and thermally drilled samples were successfully dated. The agreement of the measured abundances between both methods is a strong indication that thermally drilled samples do not suffer from systematic modern air contamination. This was an open

question because of the meltwater that fills the borehole during thermal drilling and the potential post-coring refreezing that could enclose modern air in previously open cracks in the ice. As thermal drilling offers the opportunity for more seasonally independent ice core retrieval, this outcome supports the robustness and the more versatile application of the $^{39}$Ar dating tracer.

## 5.9 Employing the time scale for interregional Ca$^{2+}$ signal comparison

By establishing a reliable age-depth scale for the WSS summit glacier, depth resolved data can now be evaluated with its corresponding age information, making the interpretation of the data set as well as the comparison with other data sets and archives feasible. Spagnesi et al. (2023) published profiles of major ions, stable water isotopes and other tracers for the WSS summit glacier, showing a distinct peak in all major ion concentrations as well as levoglucosan at a depth of 6.4 m. The detailed investigation of the WSS isotope and impurity time series warrants a separate investigation. For a first comparison with other

Alpine ice core data, we selected the Ca$^{2+}$ profile, since a connection between high Ca$^{2+}$ peaks and the occurrence of Saharan



dust transport to the Alps is well established (e.g. Wagenbach et al., 1996). Considering that a meaningful comparison with WSS requires Alpine ice core data comprising an age span starting 400 years ago at reasonable time resolution, the immediate target is Colle Gnifetti (CG) – a well-characterized site with millennial time series (e.g. Wagenbach et al., 2012).

Accordingly we compared the WSS $Ca^{2+}$ profile to a $Ca^{2+}$ record from Colle Gnifetti (Bohleber et al., 2018). Interestingly, both $Ca^{2+}$ records feature an outstanding anomaly around 1175. Applying the here derived age-depth scale to the WSS $Ca^{2+}$ data and plotting it together with the CG data in Figure 6, shows the good temporal alignment of the two anomalies within the uncertainties of the WSS time scale and not regarding the uncertainties of the CG time scale. Identifying the $Ca^{2+}$ peak with a Saharan dust event, the agreement within these two $Ca^{2+}$ records suggests that this dust event reached into both, the Western and the Eastern Alps. This implies a large Saharan dust event may have impacted large parts of central Europe at that time. On

a tentative note, aligning the major peak in both data sets would require an offset of the WSS timescale of approximately 50 a. Such a shift results also in the temporal alignment of several smaller peaks between 1550 and 1700, possibly indicating further regional-scale dust signals recorded across the Alps. The dashed line in Figure 6 shows the WSS $Ca^{2+}$ record with a 50 a offset on the WSS timescale to visualize the alignment of the peaks.

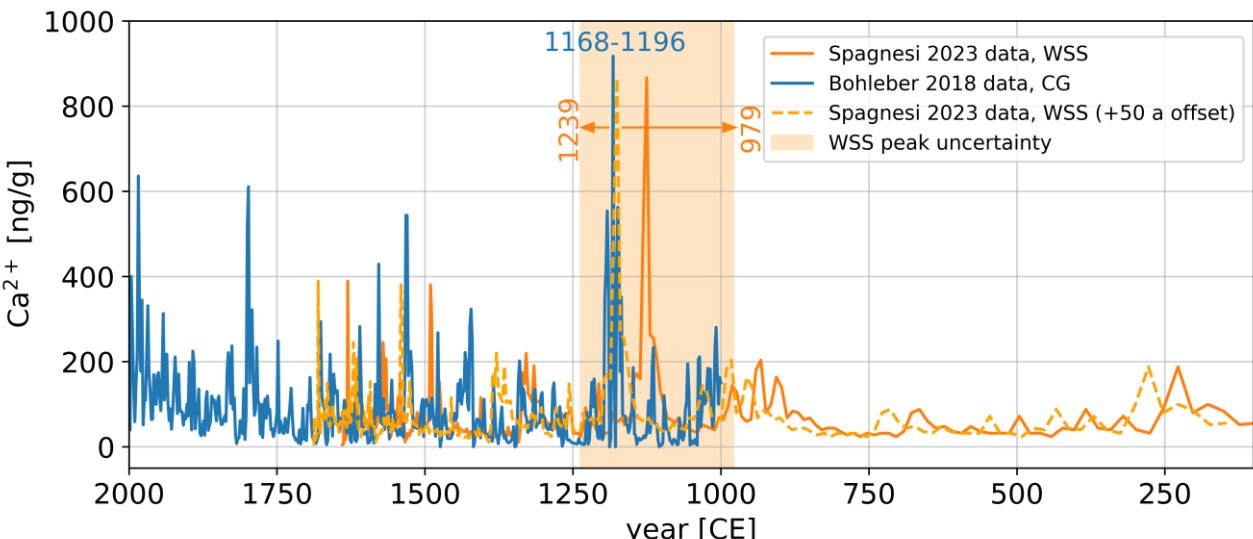

**Figure 6: $Ca^{2+}$ data from WSS (Spagnesi et al., 2023) are plotted against time using the conversion of depth to age derived in this work. $Ca^{2+}$ data from Colle Gnifetti (Bohleber et al., 2018) are plotted for comparison. The shaded area and its boundary values indicate the 1σ age uncertainty of the major peak of the WSS data set due to its time scale uncertainty. The blue values indicate the extent of the CG peak in years, not considering its time scale uncertainties. Adding a 50 a offset to the WSS timescale results in good alignment of the major peak around 1175 and also smaller peaks between 1550 and 1700.**

**6 Conclusion**

The integration of $^{39}Ar$ and $^{14}C$ ages enabled a comprehensive coverage of the remaining age profile of the WSS summit glacier through radiometrically measured ice ages. It was demonstrated that in the remaining 10 m of the glacier (as of 2019) a



continuous record of 6000 a was present, although with already ~400 a of record having been lost at the surface. Despite the loss of roughly 80 % of the original glacier profile over the past ~110 a, glaciological models were successfully applied to the age profile, showing strong agreement with the measured age data. This approach enabled the establishment of a reliable and continuous age-depth relationship for the 10 m above bedrock. The fact that the age-depth relation is continuous shows that no prolonged periods of mass loss have existed at WSS within the 6000 year glaciation history prior to today.

To derive error estimates for the model ages, MC sampling was employed, accounting for the age distributions of the measurements and as well as additional uncertainties associated with the LSS and the common depth scale. Consequently, reasonable error estimates were generated, yielding age uncertainties of 100-200 a for most of the age profile, increasing with depth. Furthermore, it was shown that different drilling methods did not influence the $^{39}$Ar age results, emphasizing the robustness of the method.

With the obtained age-depth scale, depth resolved $Ca^{2+}$ from WSS ice cores from Spagnesi et al. (2023) could now be compared more precisely to other archives, enabling broader contextualization within regional and global paleoclimatic data. A $Ca^{2+}$ record from CG published by Bohleber et al. (2018) shows a unique peak around 1175. Comparing this to the WSS data set reveals a good temporal alignment of the two signals and possibly further peaks between 1550 and 1700, making the case for period of strong, interregional Saharan dust events in central Europe.

This work shows that successfully establishing an age-depth relation for a substantially diminished glacier by the combination of $^{39}$Ar and $^{14}$C measurements has a general implication for non-polar ice core research also in other mountain regions: even under conditions of sustained mass loss and surfaces ages beyond a century, reliable time scales can be derived with the help of appropriate radiometric tracers.

**Data availability**

The probability density functions of the $^{39}$Ar and $^{14}$C measurements as well as the depth vs. age data from the 2p model, including respective uncertainties, are available at: http://doi.org/10.5281/zenodo.16528316.

**Author contribution**

DW performed the $^{39}$Ar measurements, the formal analysis and the writing. LL and JM performed the sample preparation and first formal analysis. MSW and AF were leading the drilling campaigns. AF, PB, WA and MO provided supervision, acquisition of funds, project administration and reviewed the manuscript. AS did the measurement and the analysis of the CFA data and helped with reviewing the manuscript. AJ, CK, NM and FM provided technical support in the laboratory, performed earlier work on data evaluation and helped reviewing the manuscript. TJ performed and supervised the $^{14}$C measurements at PSI and helped reviewing the manuscript.



**Competing interests**

The authors declare that they have no conflict of interest.

**Acknowledgements**

We are grateful to many colleagues for their support during the different stages of the sampling campaigns and lab work. The whole Heidelberg ATTA group is acknowledged for their support in the preparation and measurement of the $^{39}$Ar samples. For the $^{14}$C dating at PSI, Margit Schwikowski is acknowledged for the coordination of the measurements and Michelle Worek for the additional measurement in 2024.

**Financial Support**

We acknowledge financial support by German Research Foundation and the Austrian Science Fund under the DACH program: FWF: I 5246-N, DFG: AE 93/22-1 and OB 164/17-1, and by the projects Cold Ice 1 and 2: FWF P29256-N36 und FWF P34399-N.

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
