# Peer review of "A continuous 6000 year age depth relationship for the remainder of the Weißseespitze summit glacier based on 39Ar and 14C dating"

_EGUsphere, 2025_

## Referee Comment (RC1)

Wachs and co-authors present a timely study that successfully reconstructs a continuous 6000-year age-depth profile for the rapidly disappearing Weißseespitze (WSS) summit glacier. By combining $^{39}$Ar and $^{14}$C dating techniques, the authors overcome the significant challenge of unknown surface ages due to recent ablation. Different age models have been used to find the best fit to the observed data. An age-depth profile was constructed. The preliminary comparison of the $Ca^{2+}$ record with Colle Gnifetti further supported the newly constructed chronology at WSS. The paper is clearly written and the work is of great relevance to the fields of glaciology, paleoclimatology, and geochronology. I recommend this manuscript for publication, after some relatively minor revision.

**Comments and suggestions:**

First, I have some technical questions about the age models and the $^{39}$Ar measurements that I would like to discuss with the authors.

**1) Sensitivity to the Last Steady State (LSS) Assumption**

The model fitting relies on the estimation of the LSS (1914 $\pm$ 50 a, 47 $\pm$ 10 m). The uncertainty of these values is a significant potential source of error in the age model.

The author stated that the uncertainty of the LSS has been included in the MC simulation. However the data presented in the supplement material only shows the MC simulation in the p-b parameter space. In the 2p model, the ice thickness H and the accumulation rate b are correlated, meaning for a given parameter $H_0$ and $b_0$, if one chooses another $H_1$ and scales $b_1$ as $b_0(H_1/H_0)^2$ (in the case of p ~1), it will result in a same age scale with a small constant age offset. Based on this estimation, I would expect that the variation of b is at the level of ±40%, which is significantly larger than the values in Table 3. Showing the MC simulation similar to Fig. S3 in the b–H parameter space would clarify this issue.

**2) Background estimation in the $^{39}$Ar analysis**

It is a pity that the authors had to discard many $^{39}$Ar data due to various reasons. The causes given in the manuscript are certainly possible. But it just caught my eye that all the $^{39}$Ar ages despite two (WSS 24 Surface A and B) are around 400 a (see Fig. 3). Is it possible that this could be due to a residual background of the instrument? I understand that blank measurements were conducted to measure the background before the sample analysis. But were these blank measurements carried out under similar conditions to the sample measurements (i.e. gas pressure, discharge RF power, etc.)? I am asking this because the outgassing rate can be vastly different depending on the operating mode of the discharge. So, one has to make sure all measurements are carried out under similar conditions. Otherwise, the background estimated from the blank measurements may not represent the true background during the sample analysis.

**3) Discussion on Constant Accumulation Rate**

The 2p model assumes a constant accumulation rate over 6000 years. The authors have pointed out a constant accumulation rate b cannot be expected due to the changing climatic history of the

region. While the split-fit analysis (upper vs. lower samples) is a good check, the difference in values (0.71 vs. 0.46 m w.e. a$^{-1}$) warrants a more detailed discussion. The authors may consider adding some discussion about what climatic factors could cause such variability in this region and how this might affect the model's interpretation.

**Other minor comments:**

The best fit in Figure S3 is not located in the center of the red part of the MC simulations, but at the corner of it, which is a little odd to me. Could this be caused by under sampling?

What is the horizontal velocity of the glacier at the drilling position? Are there any measurements based on snow stakes? How does the horizontal movement of the glacier affect the alignment of ice cores drilled in different years?

Line 83, "…the basal layers will soon warm above…",->" …the basal layers will soon be warm above…".

Line 156, Are these counting rates for modern samples or for the sample around 400 years old?

Line 164, In the data analysis, does the rate parameter r change with time or is it set at a certain value based on the average counting rate? Could the author provide a reference containing details about the data analysis process?

Line 217, "as well as" ->"and".

---

## Author Comment (AC1)

**EGUSPHERE-2025-3681 final response to reviewers**

Dear Editor,

we appreciate the time and effort that has been put into the very constructive reviewing of our manuscript. We thank the reviewers for their input as they have raised important points and addressing them will certainly strengthen the manuscript. We carefully considered all comments and tried our best to respond to each one of them.

In the following, we provide a point-by-point response to the reviews. For some we added updated figures in order to illustrate slight changes in the uncertainties of the age-depth relationship that resulted from addressing one of the reviewers comments. Also, the corresponding data, made available at https://doi.org/10.5281/zenodo.16528315 was updated with the implemented changes. We believe that on this ground we can successfully improve the manuscript in a revision.

Sincerely,

David Wachs and co-authors

**Review #1**

*...*

*Comments and suggestions:*

*First, I have some technical questions about the age models and the $^{39}$Ar measurements that I would like to discuss with the authors.*

**1) Sensitivity to the Last Steady State (LSS) Assumption**

*The model fitting relies on the estimation of the LSS (1914 ±50 a, 47 ±10 m). The uncertainty of these values is a significant potential source of error in the age model.*

*The author stated that the uncertainty of the LSS has been included in the MC simulation. However the data presented in the supplement material only shows the MC simulation in the p-b parameter space. In the 2p model, the ice thickness H and the accumulation rate b are correlated, meaning for a given parameter $H_0$ and $b_0$, if one chooses another $H_1$ and scales $b_1$ as $b_0(H_1/H_0)^2$ (in the case of p ~1), it will result in a same age scale with a small constant age offset. Based on this estimation, I would expect that the variation of b is at the level of ±40%, which is significantly larger than the values in Table 3. Showing the MC simulation similar to Fig. S3 in the b–H parameter space would clarify this issue.*

Response: We thank the reviewer for this detailed observation. As a matter of fact there was a small inaccuracy in the text. In line 229 it should say "... were MC sampled as Gaussian distributed values with the indicated values as three standard deviations". Furthermore, the code for the MC fitting routine contained a small error regarding the consideration of the LSS uncertainties that was now corrected. This affects only the uncertainty estimation of the parameters by the MC sampling and not the best fit. While the 1sigma ranges of the fit parameters of the 2p model barely change, their distribution, given by an updated figure S3, scatters more widely:

[Figure]

Figure S3 updated: Parameter distribution of the $10^5$ MC realizations of the 2p model. The best fit was obtained by fitting the measured values, while all other realizations stem from MC sampling from the age distributions and the variation of the LSS parameters.

As the reviewer suggested, a plot of the distributions of b and H, given by figure S4, will be added to the supplementary information.

[Figure]

Figure S4: Parameter distribution b vs. H of the $10^5$ MC realizations of the 2p model. The best fit was obtained by fitting the measured values, while all other realizations stem from MC sampling from the age distributions and the variation of the LSS parameters.

Finally, as a result of the changed correlation of the fit parameters, the resulting absolute age errors increased slightly, as can be seen in an updated figure 6. These new

age uncertainties were already updated in the online repository at https://doi.org/10.5281/zenodo.16528315.
The age uncertainties might appear large, however, this was to be expected for an alpine glacier in this setting and in this progressed state of melting.

[Figure]

*Figure 6 updated: Ca²⁺ data from WSS (Spagnesi et al., 2023) are plotted against time using the conversion of depth to age derived in this work. Ca²⁺ data from Colle Gnifetti (Bohleber et al., 2018) are plotted for comparison. The shaded area and its boundary values indicate the 1σ age uncertainty of the major peak of the WSS data set due to its time scale uncertainty. The blue values indicate the extent of the CG peak in years, not considering its time scale uncertainties. Adding a 50 a offset to the WSS timescale results in good alignment of the major peak around 1175 and also smaller peaks between 1550 and 1700.*

The updated version of table 3 listing the updated 1σ boundaries of the fit parameters is below.

*Table 3 updated: Parameters for the best fit of the measured data and the 1σ boundaries from the MC fitting. Accumulation rate values are converted to meter water equivalent (m w.e.) by multiplying with an ice density of 900 kg m⁻³ (Festi et al., 2021).*

| Model | Accumulation rate, best fit, lower 1σ, upper 1σ bound [m w.e.a⁻¹] | Thinning parameter, best fit, lower 1σ, upper 1σ bound | $\chi^2_{red}$ |
|---|---|---|---|
| Nye model | **0.089**, 0.079, 0.098 | | 25.14 |
| Raymond model | **0.67**, 0.58, 0.78 | | 0.54 |
| 2p model | **0.53**, 0.38, 0.63 | **0.92**, 0.81, 0.97 | 0.4 |

**2) Background estimation in the ³⁹Ar analysis**

*It is a pity that the authors had to discard many ³⁹Ar data due to various reasons. The causes given in the manuscript are certainly possible. But it just caught my eye that all the ³⁹Ar ages despite two (WSS 24 Surface A and B) are around 400 a (see Fig. 3). Is it possible that this could be due to a residual background of the instrument? I understand*

*that blank measurements were conducted to measure the background before the sample analysis. But were these blank measurements carried out under similar conditions to the sample measurements (i.e. gas pressure, discharge RF power, etc.)? I am asking this because the outgassing rate can be vastly different depending on the operating mode of the discharge. So, one has to make sure all measurements are carried out under similar conditions. Otherwise, the background estimated from the blank measurements may not represent the true background during the sample analysis.*

Response: Thanks for raising this point! However, in this case we are very certain that the mentioned problem of a residual background did not occur. First, all measurements (reference, sample and blank measurements) were carried out under strictly equal conditions. And second, as an even more conclusive point, the measurements presented in this manuscript were conducted over the course of several months, during which the experiment was almost continuously measuring one sample per day of a large variety of environmental $^{39}$Ar samples from different origins (glacier ice, groundwater, ocean water). The measured concentrations of all these measurements scatter over the full expected range from non-detectable $^{39}$Ar concentrations (around 0 pmAr) to modern concentrations (around 100 pmAr) and therefore show that no persisting background could have led to the similar concentrations for the particular samples presented here. Also the background estimates from blank measurements were very stable over the course of the entire measurement period.

We will add a remark on the quality controls for the measurements in the revised manuscript.

**3) Discussion on Constant Accumulation Rate**

*The 2p model assumes a constant accumulation rate over 6000 years. The authors have pointed out a constant accumulation rate b cannot be expected due to the changing climatic history of the region. While the split-fit analysis (upper vs. lower samples) is a good check, the difference in values (0.71 vs. 0.46 m w.e. a-1) warrants a more detailed discussion. The authors may consider adding some discussion about what climatic factors could cause such variability in this region and how this might affect the model's interpretation.*

Response: See the response to comment 2) of review #2 for detailed reply and a rewritten paragraph. As a main reply we want to mention that the differences in fit parameter values are not significant and therefore cannot serve as the basis of a detailed discussion without being purely speculative.

*Other minor comments:*

- *The best fit in Figure S3 is not located in the center of the red part of the MC simulations, but at the corner of it, which is a little odd to me. Could this be caused by under sampling?*
- Response: Thanks for this attentive observation. We attribute this to the correlation of both parameters of the 2p model. Undersampling can be ruled out, as the behaviour does not change even for significantly higher sampling numbers. Note, however, that the offset of the best fit from the center of the red part is well within the 1sigma error of the parameters.
We will add a remark about the correlation of the fit parameters to the annotation of figure S3.
- *What is the horizontal velocity of the glacier at the drilling position? Are there any measurements based on snow stakes? How does the horizontal movement of the glacier affect the alignment of ice cores drilled in different years?*
- Response: GNSS measurements at the stake positions show that the horizontal movement is almost zero, as can be expected for the ice divide setting with the remaining relatively thin ice cover. All (repeated) point measurements of the stake positions are located within a few metres to each other and are randomly distributed within the measurement accuracy (compare Fig. 2 in Hartl et al., 2025). We will add a sentence to section 2.1 to clarify this in the manuscript. The effect of the position uncertainty of the drill locations on the depth scale alignment of the ice cores is discussed in lines 94-100 in the manuscript.
- *Line 83, "…the basal layers will soon warm above…",->" …the basal layers will soon be warm above…".*
- Response: Thanks for observation. We suggest to rephrase: "… the basal layers will soon rise above 0°C in temperature…"
- *Line 156, Are these counting rates for modern samples or for the sample around 400 years old?*
- Response: Good point, this was not immediately clear from the text. We suggest to rephrase: "With usual count rates of around 6 $^{39}$Ar atoms h$^{-1}$ for modern samples, sample measurements…"
- *Line 164, In the data analysis, does the rate parameter r change with time or is it set at a certain value based on the average counting rate? Could the author provide a reference containing details about the data analysis process?*
- Response: As the background increases over the measurement time due to outgassing from the walls, the rate parameter r (giving the frequency of atom detection events) increases over the measurement time. It incorporates, however, the measured background build-up from blank measurements. The details of the data analysis routine will be published elsewhere.
- *Line 217, "as well as" ->"and".*
- Response: thanks, will be corrected.

**Review #2**

...

*However, some aspects need corrections or adjustments:*

*1) The authors maintain that the data presented allow us to assert that there is no hiatus over the upper 2000 years of the profile. In Figure 4b, the zoom on the dated points, this conclusion is not obvious. In my opinion, between the $^{39}$Ar and $^{14}$C point series, it seems to me that these two series are not as well 'aligned', despite the uncertainty bars. This could possibly be due to a hiatus during this time interval, or as proposed that the $^{39}$Ar dates are inaccurate due to the half-life value not being updated.*

Response: The reviewer touches upon an interesting point. We tried to be very careful in our wording ("At least for the past 2000 a, a hiatus in the order of 400 annual layers seems to be unlikely given the good fit of the model to the data with errors of measurements and fit well below 400 a", lines 300-302) because the natural question would of course be: what number of annual layers do we mean when we speak of a hiatus? We certainly do not claim to be able to detect hiatus of a few years and probably this is not the point of critique as claiming this would be unrealistic given our methods and also not of interest in the presented context. With the uncertainties presented in this manuscript, only hiatus of several hundred years are at all detectable, which is why we speak of "hiatus in the order of 400 annual layers".

Concerning the argument that the $^{39}$Ar and $^{14}$C point series are not well aligned, we agree that it visually appears like that in figure 4b. However, this impression originates from the uppermost $^{14}$C data point, as far as we see it, and this one exhibits very large errors. Claiming a misalignment based on this one data point is, in our opinion, an overinterpretation of this one measurement. With our wording that a hiatus "seems unlikely" we tried to phrase it in a way compatible with these uncertainties.

Concerning the new $^{39}$Ar half-life of 302 a, we perform the comparison of an analysis based on this "new" value with the currently used one in section 5.7 and even comment on the better fit quality due to a slightly better agreement of the $^{39}$Ar and $^{14}$C age values. However, given the uncertainties, this is by no means so conclusive as to rule for or against one or the other half-life value. Nevertheless, also addressing the comment of the reviewer made in the special remarks section ("*Line 318-328: Should the new estimate of the half-life of $^{39}$Ar not be used, or at least compared with the current value, in this work?*"), we will add corresponding Figures to Figures 4 and 5 with the new half-life in the supplementary information part.

*2) The authors conclude that the combination of $^{39}$Ar and $^{14}$C has provided a significant advance in dating by allowing a better fit of the best dating model. For this conclusion, it would have been interesting to be able to judge this by presenting the best model fit using only the $^{14}$C and glaciological data, then a new version adding the $^{39}$Ar dating points, even if a short test (line 305-309) tries to convince us.*

Response: This point touches upon a similar topic raised by the first reviewer in comment 3 and raised again with a slightly different focus in comment 4 of this review. Certainly we have to explain ourselves better here, since we were thinking about exactly the raised questions during the writing process of this manuscript, but seemingly did not yet succeed in understandably explaining our reasoning. The first raised point aims at clarifying the benefit of adding $^{39}$Ar dating to the already existing $^{14}$C point series. The second point asks for a more detailed interpretation of the fit parameters in the glaciological setting.

For both points, the paragraph from lines 305 to 310 is the one that is in question. In the following we will propose a rephrased version for this paragraph which hopefully conveys the information more concisely. The paragraph also contains the slightly updated parameter values given in the updated version of table 3 in this document.

*"The models employed in this work are certainly simplifications. However, in order to draw conclusions from the limited data set, the complexity has to be reduced by a model. By the successful application of the 2p model in this work, we are able to show that we can generate a good fit that is consistent with the hypothesis of no hiatus based on a simplifying assumption. This simplifying assumption is that of a constant net accumulation rate and thinning parameter over the whole profile of the ice column. In order to assess if this assumption is tenable, we performed a fitting of the model with only the lower six sample ages, which represent all $^{14}$C samples and compare them to the full fit. It results in an accumulation rate parameter of $b_{lower}$ = 0.42 (1σ interval: 0.30 to 0.56) [m w. e. a-1]. Comparing this to the original value of b=0.53 (1σ interval: 0.38 to 0.63) [m w. e. a-1] ascertains that they only deviate within their 1σ intervals and therefore the difference is not significant. The same applies to the thinning parameter ($p_{lower}$ = 0.84; 1σ interval: 0.72 to 0.93).  This result gives confidence that the net accumulation rate did not significantly change between the older and younger parts of the ice profile, at least to the degree to which we can resolve it. Thus, the 2p model, despite its simplifying assumptions, does not lead to a problematic oversimplification and is able to capture the important aspects of the setting at the WSS summit glacier. The interpretation of this seemingly stable net accumulation rate in terms of climatic factors at this location is not trivial and can hardly be done in a non-speculative way with the data at hand. It will have to be done elsewhere in conjunction with more data.*

*A second conclusion can be drawn from comparing the purely $^{14}$C data based model analysis with the one containing all $^{14}$C and $^{39}$Ar sample ages. The consistency between*

*the fit results ascertains the compatibility of the dating techniques and how [39]Ar data enables to determine the younger parts of an age-depth model based on data and not only as a pure extrapolation of the older part."*

Furthermore, we suggest to rephrase lines 278-279: *"Therefore, the result should be understood as a net accumulation rate parameter that gives a rough estimate of the long-term net accumulation rate average over the last six millennia. It should be noted that this does not trivially translate into a precipitation rate or other climatic factors".* This way we underline our finding of a net accumulation rate, which does not trivially translate into the annual precipitation rate.

*3) On these summit glacier sites, Weißseespitze, Colle Gnifetti or Dôme du Goûter, the net accumulations are not representative of precipitation because a major part of the winter and/or annual snow is blown away. This is one of the reasons why these sites are unique and have preserved fairly old ice, which is interesting for certain studies. On the other hand, this means that the annual record is not complete (often only wet summer snow is accumulated). Under these conditions, the comparison of chemical profiles, where the aim is to identify specific events and compare them with other sites, is questionable. The dust layers may have different sources, Saharan dust, or a longer period of local dust deposition following a period of ablation. In this manuscript, I propose to delete the paragraph concerning the Calcium profile and their comparison with Colle Gnifetti.*

Response: Thanks for pointing this out and certainly the reviewer touches upon an important point here. We respectfully argue, however, to leave this section in the manuscript. Section 5.9 intends to illustrate how, e.g. under which limitations, the derived age-depth relation can be applied to other data and how important it is to have reasonable age uncertainty estimates. This is an important message of the manuscript and we would thus like to keep it. For this purpose, it was the obvious way to compare a record from WSS with another alpine record which stretches this far back and is well dated. Comparing the Ca profile as a measure of Saharan dust is an easy check in this context and the very clear signal peaks comprise a fairly simple setup. The detailed discussion of the shortcomings of alpine glacier sites in preserving chemical profiles is beyond the scope of this manuscript, but we now note this limitation more explicitly following the reviewer's comment. We, therefore, suggest, that we include the raised critique as an additional sentence and point out the exemplarity of the comparison. Proposed additional sentence at the end of line 348: *"The comparison is done as an example of how the derived age-depth relation can be applied to the interpretation of e.g. chemical profiles. A detailed interpretation of the time series with regard to post-depositional effects or the seasonality of the accumulation, however, is beyond the scope of this example."*

*4) With regard to the accumulation values obtained by the models, it is clear that these have not been constant over this long 6000 years period. First of all, the values obtained with the 6 high points and then the 6 low points are quite different. Can you either expand on your conclusions, or use a model that allows accumulation to evolve over time?*

Response: The response to this comment is included in the response to comment 2) of review #2. As the fit parameters agree within their 1sigma range, they do not significantly deviate from each other and we therefore do not interpret it further in this direction.

*Special remarks:*

- *Lines 24-25: "novel combination" is not correct, it has been used and published previously (Tibet, Dôme du Goûter).*
- Response: thanks for pointing this out, the word novel will be erased and the reference ( Shugui Hou et al., A radiometric timescale challenges the chronology of the iconic 1992 Guliya ice core.Sci.Adv.11, eadx8837(2025). DOI: 10.1126/sciadv.adx8837) will be added.
- *Line 45: add significant references to Alpine Holocene records.*
- Response: we propose to add the following references:
  - Jenk, T. M., Szidat, S., Bolius, D., Sigl, M., Gäggeler, H. W., Wacker, L., Ruff, M., Barbante, C., Boutron, C. F., and Schwikowski, M.: A novel radiocarbon dating technique applied to an ice core from the Alps indicating late Pleistocene ages, Journal of Geophysical Research: Atmospheres, 114, https://doi.org/10.1029/2009JD011860, 2009
  - Bohleber, P., Erhardt, T., Spaulding, N., Hoffmann, H., Fischer, H., and Mayewski, P.: Temperature and mineral dust variability recorded in two low-accumulation Alpine ice cores over the last millennium, Climate of the Past, 14, 21–37, https://doi.org/10.5194/cp-14-21-2018, 2018
  - Legrand, M., McConnell, J. R., Preunkert, S., Wachs, D., Chellman, N. J., Rehfeld, K., Bergametti, G., Wensman, S. M., Aeschbach, W., Oberthaler, M. K., and Friedrich, R.: Alpine ice core record of large changes in dust, sea-salt, and biogenic aerosol over Europe during deglaciation, PNAS Nexus, 4, pgaf186, https://doi.org/10.1093/pnasnexus/pgaf186, 2025
- *Lines 74-75: add information on temperatures, figures or data.*
- Response: The data on ice temperature has been published in the referenced sources. We will add the following red part to lines 74-75 to provide more information: *"Measurements of borehole temperatures indicate that the ice is still frozen to the bedrock with mean annual ice temperature at the lowest sensor (~1m above bedrock) at around −3.3 °C and maximum values not exceeding −2.6 °C (Bohleber et al., 2020; Fischer et al., 2022; Stocker-Waldhuber et al., 2022b)."*

- *Line 76: "is rapidly declining", add data*
- Response: We will add a reference to table 1 to the sentence. The rapid decline is also further elaborated on in lines 80-81.
- *Line 97-100: If the glacier is covered by a homogeneous layer of dust on the surface, this does not necessarily mean that it comes from a deep layer that appeared following the melting of the surface, but it may come from deposits of recent particles accumulated over the entire ablation period, just hidden during the winter but returning the following spring.*
- Response: Thanks for pointing this out. However, the dust layer had already been visible in the 2019 ice cores and definitely melted out of the ice profile (compare Fig. 1 in Fischer et al., 2022).
- *Line 125: Illustrating the 3H profile in Figure 3 could complement the data presented.*
- Response: The information of the tritium measurements, which were published by Bohleber et al., 2019, is limited to the information that no post-1963 ice remained in the ice column. We fear that adding this information as minimum age markers to Figure 3 would only add very low information content but could greatly reduce the clarity of the plot.
- *Line 179: Why has a $^{14}$C value been removed?*
- Response: I think the reviewer misunderstood the sentence. We meant that in 2024 an additional sample was measured that is considered in our analysis, which, however, naturally was not part of the 2019 publication.
- *Line 305-308: Do the 6 upper dots represent $^{39}$Ar, and the 6 lower dots $^{14}$C?*
- Reponse: Good point, we consider 11 sample ages in our analysis. The five upper ones based on $^{39}$Ar analysis, the lower 6 based on $^{14}$C analysis. In the rewritten paragraph proposed in response to comment 2) of review #2, we have removed the analysis of only the upper 6 data points (all $^{39}$Ar sample ages plus the uppermost $^{14}$C sample age), as this only adds confusion to this already complex section.
- *Line 318-328: Should the new estimate of the half-life of $^{39}$Ar not be used, or at least compared with the current value, in this work?*
- Response: We do not fully understand the question of the reviewer. The mentioned section (lines 318-328) serves exactly that purpose, the comparison of the newly proposed $^{39}$Ar half-life with the currently used one. However, as the results do not deviate significantly, we do not feel the need for a more in-depth discussion of this matter. We will, however, add figures to the supplementary information which correspond to figures 4 and 5 from the main text with the analysis done with the new half-life estimate.
- *Line 339-364: The discussion of calcium profiles is not convincing.*
- Response: Please refer to the response to comment 3) of review #2.